# Economic violence against women: A case in Turkey

**Ömer Alkan**[1¤a]*, **Şenay Özar**[2☯¤b], **Şeyda Ünver**[1☯¤a]

**1** Department of Econometrics, Ataturk University, Erzurum, Turkey, **2** Department of Economics, Van Yuzuncu Yil University, Van, Turkey

☯ These authors contributed equally to this work.
¤a Current address: Department of Econometrics, Faculty of Economics and Administrative Sciences, Ataturk University, Erzurum, Turkey
¤b Current address: Department of Economics, Faculty of Economics and Administrative Sciences, Van Yuzuncu Yil University, Van, Turkey
* oalkan@atauni.edu.tr

**Data Availability Statement:** The data underlying this study is subject to third-party restrictions by the Turkey Statistical Institute. Data area available from the Turkish Statistical Institute (bilgi@tuik.gov.tr) for researchers who meet the criteria for access to confidential data. The authors of the

## Abstract

The aim of this study was to determine the factors affecting the exposure of women in the 15–59 age group in Turkey to economic violence by their husbands/partners. The micro data set of the National Research on Domestic Violence against Women in Turkey, which was conducted by the Hacettepe University Institute of Population Studies, was employed in this study. The factors affecting women's exposure to economic violence were determined using the binary logistic regression analysis. In the study, women in the 15–24, 25–34 and 35–44 age group had a higher ratio of exposure to economic violence compared to the reference group. Women who graduated from elementary school, secondary school, and high school had a higher ratio of exposure to economic violence compared to those who have never gone to school. Women's exposure to physical, sexual and verbal violence was also important factor affecting women's exposure to economic violence. The results obtained in this study are important in that they can be a source of information for establishing policies and programs to prevent violence against women. This study can also be a significant guide in determining priority areas for the resolution of economic violence against women.

## Introduction

One of the most common forms of violence against women is violence perpetrated by a husband or other male partner. Intimate partner violence (IPV), often referred to as domestic violence, takes several forms [1]. An intimate partner is the partner/companion with whom the woman has sexual intercourse, or the father of the child she carries [2]. In the literature, it is stated that there are four different types of non-physical violence: emotional, psychological, social and economic violence [3]. Physical and sexual violence, however, refers to the type of violence related to physical intervention against women [2, 4]. IPV is a comprehensive and multi-directional social problem associated with various health and social consequences. IPV

study did not receive any special privileges in accessing the data.

**Funding:** The author(s) received no specific funding for this work.

**Competing interests:** The authors have declared that no competing interests exist.

against women is a common and significant public health issue [5–7]. Globally, it is estimated that one in three women have been subjected to intimate partner violence at some point in their lives, but these estimations vary widely between countries [8–11]. Violence against women is considered a serious violation of human rights [12–14]. The fact that intimate partner violence is a major social issue affecting a large number of women and children is now undeniable [15]. Partner violence cannot be interpreted without taking into account the significant differences between the drives of the perpetrators, the types of violence, the social positions of the woman/man and the cultural situations in which the violence occurs [16].

One domain of intimate partner violence is economic violence [17]. Economic violence is often considered within the scope of emotional or psychological violence [3, 18]. But recently, scholars have begun to define economic violence as a unique form of violence [3]. Economic abuse is a unique and mandatory form of control behavior that the abuser uses in an intimate relationship other than physical, sexual and psychological abuse [19]. Since economic violence is an important aspect of IPV, studies that ignore economic violence miss an important factor [20]. Briefly, economic violence is a field of research that has emerged recently [21].

In order to develop a more comprehensive assessment of the experience of economic violence, scholars have begun to develop economic violence scales and sub-scales [3]. First, the two sub-dimensional (economic control and economic exploitation) Scale of Economic Abuse (SEA) was developed to measure economic violence [22]. Later, this scale was revised, and three sub-dimensions (economic control, employment sabotage, and economic exploitation) were added to the Scale of Economic Abuse (SEA-12) [23]. The validity and reliability of SEA-12 were also tested [24]. In another study conducted in China, the Chinese version of the SEA-12 was adapted, and its validity and reliability were tested [19].

## Definition of economic violence

The term"economic abuse" first emerged in the late 1980s [5]. Economic violence involves controlling a woman's ability to gain, use and sustain economic resources, thereby threatening her economic security and potential of self-sufficiency [3, 17, 18, 22, 23, 25]. Economic abuse can a very powerful tactic in manipulating, dominating and controlling a person for the purpose of encouraging dependence or abusing them financially [21]. Economic violence focuses on creating economic dependence on the perpetrator. It occurs when the victim's financial resources are under full control [3]. It is the abuser's having full control on the victim's money and other economic resources or activities [26]. Economic violence against women is characterized by male partners, who have absolute control over financial resources, keep financial resources or refuse to contribute financially to their female partners, thus leading women to complete dependence for their most basic needs and satisfaction [27].

There are many types of economic violence against women. Tactics such as intervention at work, preventing the spouse from working outside of the home or the community, harassing or disturbing the spouse in her workplace, preventing or limiting education, regulating access to money or refusing access to financial information, stopping or restricting funds necessary for needs such as food and clothing, stealing money from spouse, refusing to work and the creating debt on the part of the woman, dominating family economy by making unilateral decisions, ruining the credit note of woman on purpose are economic violence behaviors [18, 24–26, 28–30]. Behaviors such as taking jewelry given to woman at the wedding ceremony and asking for bride price and dowry are also considered as economic violence [28]. These strategies are used by men to maintain economic control and assert their dominance in the domestic environment, putting women in secondary positions [31]. These tactics may include harming victims' economic self-sufficiency and self-efficiency [4].

## Consequences of economic violence

Economic violence can seriously impede women's economic, physical and psychological health [22]. Exposure to economic violence is related to public health, as it threatens the economic security and independence of the victim, limits the capacity to leave abusive relationships, and potentially leads to adverse mental health effects [25]. Economic violence also has negative consequences in terms of women's safety. Limiting women's financial resources leads to financial dependence on the abuser, which often traps women in abusive relationships [18]; this economic dependence on the abuser is a direct consequence of economic abuse. This, in turn, poses a critical obstacle for many women who try to leave their abusive partners [32].

According to marital dependency theory and interdependence theory, women who are forced into economic dependence are at risk of being stuck in a relationship. Women's economic concerns are regarded as one of the main reasons why they have difficulty leaving a violent relationship [23]. In fact, economic violence reduces trust and women who are not currently married but are living with a male partner may be less likely to take these steps [20, 33]. Interviews with victims of violence have generally shown that victims' experience of non-physical forms of violence is equal to or more damaging than physical violence [34]. Even after the end of economic violence, women's health is severely affected [7].

There is substantial evidence showing that gender-based violence (GBV) negatively affects women's physical and mental health, causing headaches, physical injuries, eating problems, stress, fear and anxiety, sleep and other problems [32]. Like all other intimate partner abuse and violence, economic abuse is a social problem as well as a significant element of a personal relationship [21]. Economic violence is associated with depression and anxiety [4]. Higher levels of economic violence are associated with a greater increase in depression [30, 35–37]. Economic violence plays an important role in the psychological well-being of IPV victims; thus the inclusion of economic violence in the measurement of IPV is beneficial [5]. Economic violence has harmful consequences for the economic and psychological well-being of victims. Economic violence has recently been associated with increased symptoms of depression and anxiety and decreased quality of life [5]. Economic violence affects psychological and physical health through the stress associated with poverty and the facing of an uncertain financial future [29]. Economic abuse has a negative effect on women's cardiovascular, psychosocial, and overall health [31]. It is associated with economic abuse, financial difficulties, psychological problems, and depression. Financial difficulty and dependence represent significant obstacles to women leaving violent relationships [21]. Women who have experienced emotional or economic violence without physical or sexual violence in the past are more likely to report symptoms such as anxiety or grief, sadness due to feelings of worthlessness, wanting to cry for no reason, mood swings, bad temper, insomnia and persistent fatigue than women who have not reported IPV throughout all their lives [7].

Economic violence negatively affects women's economic well-being [18]. Economic violence leads to increasing poverty due to women's reduced access to sources of independent living [26]. It is important to note that the relationship between poverty and economic abuse is complicated [37]. Poverty and socioeconomic inequality are both causes and consequences of economic abuse [26, 38]. Poor women are more likely to be dependent on their male partners, and such dependency can be used as a tactic to control women, and this situation may lead to abuse [37]. Economic violence is a very powerful and deadly form of abuse. It is also a form of discrimination against women [26]. Economic abuse can result in rising poverty, a known and significant indicator of poor health among women in all populations. Poor women have limited life options and can be forced to live in environments that increase their vulnerability to diseases [31]. Economic abuse can also indirectly affect women's physical and psychological

health. Studies have shown a strong relation between poverty conditions and poor physical and psychological health [22]. Economic violence can also create an obstacle to women's physical or mental health if the economic resources that a woman personally creates are consumed or abused by the actions of her partner [39]. Economic coercion has traumatic effects on women including feelings of, humiliation and a distorted sense of self, and it results in in limited opportunities and fewer chances for a better quality of life [10].

Another consequence of economic violence is that it causes social inequality and encourages sexual abuse of girls/young women by older men. It creates high demand for commercial sex by relatively rich men and a desire to break the poverty cycle of young women by any means necessary; therefore women may commercialize their bodies as a means of rapid enrichment [26].

## Relationship of economic violence with other types of violence

Economic abuse can increase the risk of other forms of violence, including physical, emotional, and sexual violence. Scientific studies show strong positive relationships between economic abuse and both physical and emotional violence [37, 38, 40]. Economic abuse tends to occur along with other forms of violence and may coincide as part of controlling behavior [38]. It has been found that women who are exposed to economic violence are also more likely to be affected by other types of violence (psychological, physical, or sexual) [39]. Physical IPV and emotional IPV are effective on economic violence [36]. Different forms of violence are mostly intertwined rather than single events and are continuous, and they form a "systemic violence" [10]. Economic abuse can be equally severe, with significant consequences on the health results of victims [31].

Economic violence are different from other forms of IPV (physical, psychological, and sexual violence), but is moderately associated with them [17, 39–41]. Victims who are exposed to a form of violence (physical, psychological, or sexual) are probably exposed to economic violence as well [5, 35]. Women are exposed to economic violence and physical violence more than men [24]; in addition, the risk of those who are exposed to economic violence to be exposed to physical violence are higher than those who have not experienced economic violence [34]. Economic abuse is more common among women who experience IPV and activity restriction [36]. It can also indirectly affect women's health through its association with economic, physical, sexual and emotional abuse [31].

## Economic violence and gender roles in Turkey

Violence against women, which is serious global problem, is also one of Turkey's important social problems. All women around the world face the risk of being exposed to gender-based violence, regardless of country, ethnicity, class, religion, economic and/or social status. For this reason, the struggle against violence towards women has not only taken place within the borders of nations but has also gained an international dimension [13]. Different studies conducted in Turkey in recent years show that one in three women have been exposed to violence at some point in their life. Although the lack of legal data on violence against women results in limited information on whether violence against women is increasing or not, high figures show that this is a multidimensional social problem fed by structural dynamics [9].

Economic violence in Turkey also varies by profession and region. It has been found that 10% of the participants in the study conducted with nurses in Şanlıurfa [42], 1.7% of the participants in the study conducted with general practitioners [43], and 13.6% of the participants in the study conducted with women working at Sivas Cumhuriyet University were exposed to economic violence [44]. In a study conducted in Manisa, it was found that 11.2% of women

were exposed to economic violence in the last 12 months [45], in another study conducted in Manisa 7.4% of women [9] and again in another study conducted in the Manisa city center, 24.4% of women were exposed to economic violence [46]. It was found in a study conducted in Konya, that 13.5% of married women were exposed to sexual and economic violence [14]; in another study conducted in Turkey's capital Ankara showed that 60.4% of women [28] and 19.3% of the women in Edirne, were exposed to economic violence [47]. In a study conducted with women admitted to the IVF Center in southwest Turkey, 19% of the participants [48] and, in another study conducted with individuals aged 65 and over living in Çanakkale, it was found that 12.2% of the participants were exposed to economic violence [49].

Domestic violence against women is a major social problem arising from unequal power relationships between men and women [45]. Turkish society has a traditional, patriarchal structure in which the culture has different hierarchies and different areas of activity for men and women [50]. Violence perpetrated by intimate partners is often used to demonstrate and strengthen a man's position as head of the household or relationship [1]. That a man's control over his wife is an important part of masculine identity is part of a culture of honor in Turkey [51] were, men are allowed to dominate women. The role of the husband is authoritarian, and the husband has the right to use all means to support his family, while the role of the woman is to look after family members and her husband [52]. In patriarchal societies, such as Turkish society, it is believed that the economic management of the family is in the hands of men. This belief can result in a higher rate of economic abuse [42]. A broad spectrum of literature on Turkey provides evidence that social norms related to gender roles have a significant effect on society, especially women. Traditional social norms regarding gender roles have a growing effect on IPV [53]. Social norms in Turkey also affect attitudes towards spouse abuse. In Turkish culture, spouse abuse is considered acceptable and a private family matter that should not be discussed with others [50]. Women want to hide the violence they are exposed to, as it is thought that these should stay in a more private space [54]. Statistics reflecting a high level of violence in Turkey reveals that the problem is a social problem fed by gender inequality, socio-economic situation and patriarchal cultural bias [55].

The gender ideology of Turkey can be understood by looking at the state of women in education and employment, their positions inside and outside home, and observing how these positions reflect social practices [56]. Like any socio-cultural structure, Turkish society also has its own social values. In the Turkish socio-cultural structure, there are social values that teach and legitimize violence and make people desensitized towards it. For instance, taking over a woman's income is also expressed by some women as a necessity of being a family, which does not qualify as violence [46]. Social norms also affect the division of labor between men and women in Turkey. For example, men are expected to be responsible for farm-related tasks, physically heavy work, and outside relationships. On the other hand, women are responsible for housework, gardening, and pet and child care. Moreover, it is considered shameful that men do "women's work," affirming claims of hierarchy between masculinity and femininity [53]. The main reason that violence against women has been allowed to increase has been proven to be the great imbalance of power and control that exists between men and women in a society strongly affected by the patriarchal worldview [10].

Little is known about economic violence in Turkey. Research on economic violence against women in Turkey is very limited. As far as we know, this is the first study known to determine the factors affecting economic violence against women in all around Turkey.

In this study, the research questions focused on the state of economic violence of women living in Turkey are as follows:

Research Question 1: To what extent do women suffer economic violence?

Research Question 2: How are the women in this study experiencing economic violence
according to various factors?

Research Question 3: Do sociodemographic and economic conditions of women affect the
state of economic violence?

Research Question 4: Do partner-related factors have an effect on women's exposure to eco-
nomic violence?

Research Question 5: Is there a relationship between economic violence and other types of
violence?

## Literature review

Among non-physical types of violence, economic violence, despite its potential significance,
has not been adequately studied globally. Until now, limited empirical research has been con-
ducted on the links and consequences of economic violence [4, 18, 25, 39, 57]. In recent years
scales have been developed, tested and revised by scholars to measure economic violence [5,
17–19, 22–24].

Looking at the studies conducted, the prevalence of economic violence varies between
countries. In a study conducted in the United States, 94% of women who experienced violence
from their intimate partners [24], 11% of women in a study conducted in Vietnam [25], and
about 12% of women in a study conducted Lebanon were detected to have been exposed to
economic violence [32]. Also in the United States in another study conducted with university
students, it was determined that 77% of those who reported any IPV experience (physical, sex-
ual, or psychological abuse) were exposed to at least one experience of economic violence [17].
It was found in a study conducted in Poland that 8.8% of the participants were victims of eco-
nomic violence [58], 89% of the participants in a study conducted in Kyrgyzstan experienced
at least some form of economic abuse [40], 19% of the women in a study conducted in Croatia
were exposed to economic violence [59], and it was found that 50.4% of the men in a study in
Laos perpetrated economic violence against their spouses [29]. 27% of pregnant women in a
study conducted in Ethiopia [2], 52% of women in a study conducted in Ghana [60], 10.8% of
women in a study conducted in Spain [7] and 18.9% of women in a study conducted with refu-
gees in Italy were found to have been exposed to economic violence [61]. In a study conducted
in the Philippines, it was stated that the prevalence of economic abuse ranged from 1.5%
(income controlling by the spouse or forcing women to work) to 6.9% (loss of job/income
source due to the husband) [38]. In a study conducted in Australia, it was found that preva-
lence of economic abuse over a lifetime was 11.5% [36]. In another study conducted with
micro-entrepreneur women in Peru, it was found that 22.2% of those women were exposed to
economic violence at some point in their lives and 25% were forced to take bank loans by their
partners against their will [39].

Women's exposure to economic violence is associated with many factors. Women with low
levels of education were found to be more likely to suffer economic abuse [36, 39, 60]. In con-
trast to these studies, another study also states that compared to women who attended fewer
years of schools, those who attended more years of school are just more likely to report eco-
nomic violence [25]. The partner's level of education is also effective on economic violence
[49]. Educational differences between of the woman and her husband/partner are also signifi-
cantly associated with economic abuse [38].

Economic violence is linked to low income [36, 39, 44, 45, 47]. Having a higher standard of
household living reduces the likelihood of economic violence and any IPV occurring together

[25]. It has been found that a significant proportion of women who report economic violence from their spouses have a lower family income and those who are non-working women than those who do not experience violence [62]. Factors such as financial stress and financial resilience are also associated with economic violence [36]. A study conducted with older adults found that older adults who were financially independent are less likely to suffer psychological and economic violence [49]. In addition, the income differences of the woman and husband/partner are significantly associated with economic abuse [38].

The labor situation variable has an effect on economic violence [36]. Unemployment is significantly associated with forms of economic abuse [38]. Lack of economic independence is an effective factor for economic violence [49].

There is a significant correlation between the age of women and their exposure to economic violence [36, 44]. The age of the partner is also significantly associated with women's IPV [2]. In particular, it was found that women who reported economic abuse by an intimate partner were older and younger women were less likely to report economic abuse [25, 40]. In a study conducted, experts stated that young adults experience economic abuse, negative economic conflict, and economic control. Experts stated that more work needs to be done to improve the financial literacy of young adults [21].

It has been found that the marital status variable has an effect on economic violence [36]. Singles are less exposed to economic violence than married people [49]. Living with a partner/partner is a factor associated with economic violence [39].

There is a significant relationship between economic violence and the mother's history of violence [42]. Women who witness IPV against their mothers are more likely to report economic violence and any IPV at the time of occurrence [25]. Variables of parental violence in terms of the husband are associated with economic violence [45]. Those who have been exposed to domestic violence by family members in their childhood perceive domestic violence as normal in their future lives or marriages [14]. Childhood sexual abuse exposure is associated with economic violence [63]. Having a history of child abuse also increases the likelihood of being exposed to sexual and economic abuse [6].

There is also a relationship between the economic abuse of the partner and the state of parenthood. In particular, women who do not have children are less likely to report economic abuse by their partners [40]. Women who had more live births are marginally less likely to report only economic violence than women who had fewer live births [25]. Women who report economic abuse by an intimate partner are more likely to have children [39, 40]. Older people who live with their children are exposed to economic violence significantly more than those who do not [49].

There is a meaningful relationship between the structure of the women's family (number of households, making decisions, etc.), the socio-cultural and economic structure in society and exposure to economic violence [44]. It has been found that increased socio-cultural cohesion significantly reduces the likelihood of women being exposed to emotional, sexual and economic abuse and to two or more types of abuse [6]. Economic decision-making and family planning decision-making variables are factors associated with economic violence [60].

The health status of men also affects economic violence against women [29]. Disability status and health status variables have an effect on economic violence [36]. Partners with aggressive behavior are significantly associated with the IPV of pregnant women [2]. The husband's gambling habit is also associated with economic violence [47]. Alcohol use by the intimate partner is also significantly associated with pregnant women's IPV [2]. Alcohol use variables of women and spouses have been found to be effective in economic violence [60]. Chi-square analyses have shown a significant positive association between recent injection drug use and economic abuse of a non-intimate partner [40]. There is a significant relationship between

economic violence and a smoking habit [42]. The history of partners who smoke is significantly associated with the IPV of pregnant women [2].

Another study also showed remarkable results. It points to the conclusion that the intersection of Orthodox neoliberal politics and privatization with patriarchy, nationalism and conflict encouraged economic violence against women in the region [57].

Unfortunately, violence has extremely long-lasting effects, even when women are no longer exposed to abuse. There are also generational reflections of violence, including economic violence [26]. In addition, economic and psychological violence have long-term effects on mothers' depression and parenting [64].

## Materials and methods

### Data

The study used cross-sectional data of the National Research on Domestic Violence against Women in Turkey conducted by Hacettepe University Institute of Population Studies in 2008 and 2014.

In order to understand and determine the causes of the different dimensions of violence against women and to fulfill the need to collect data on this subject, in 2008, a comprehensive the National Research on Domestic Violence against Women in Turkey was held for the first time. The National Research on Domestic Violence against Women in Turkey, which was carried out in 2014, is important in terms of reflecting violence against women during a time of change. The National Research on Domestic Violence against Women in Turkey is one of the most comprehensive studies conducted nationwide in order to understand the extent, content, causes, and consequences of domestic violence experienced by women, as well as the risk factors [13, 65].

As part of the Violence Survey, Turkey is divided into 30 layers to provide estimations at the level of country, urban/rural, 12 regions, and five regions. Except for the Istanbul region which is one of the 12 regions, approximately 75 percent and 25 percent distribution was made between the urban and rural layers. In Istanbul, about five percent of households were selected from the rural layer. In the study, settlements with a population of 10,000 and above constituted urban areas, and settlements with a population of less than 10,000 constituted rural areas. The sample type of the research was cluster sampling [13, 65].

In a 2008 study, 12,795 women were interviewed face-to-face, and the female question paper was completed with a rejection rate of 2.1%. The answer rate in the female interviews was 86.1% [65]. In the 2014 survey, 7,462 women were interviewed face-to-face and the female question paper was completed, with a rejection rate of 4.4%. The answer rate in the female interviews was 83.3% [13]. These data sets included female weights calculated to match the sample design of the study [13, 65].

In this study, women who were married, had a relationship or had any previous relationship were included in the analysis. Single women who had never been in a relationship before were not included in the study. In conclusion, the data of a total of 18225 women aged 15 years and over who participated in the National Research on Domestic Violence against Women in Turkey, including 11514 women in 2008 and 6711 women in 2014, were employed.

### Outcome variable

The question paper for the study of the National Research on Domestic Violence against Women in Turkey was designed by taking into account the question papers used by the World Health Organization's Multi-country Study on Women's Health and Domestic Violence Against Women [13, 65]. In the National Research on Domestic Violence against Women in

Turkey, women were also asked three questions to determine their experiences of economic violence throughout their lives. Questions including, "Has your spouse or any of the people you have been with prevented you from working or forced you to leave a job against your will?" "Have they ever refused to give you money for you to fulfil the needs of the household even though they have enough money for some other expenses?" and "Have they ever tried to take your own money (in a situation where you have an income) against your own will?" were asked. There are also studies that use these three questions to measure economic violence [25, 60]. The term husband/partner in this study also includes 'never married' women who had or were in a cohabiting relationship with an intimate partner to whom they are or were not married. As the study is related to the economic violence of the husband/partner against women, women who stated in the surveys that they had never married or had never had a relationship in the present/past were excluded from the analysis.

Subcategories of economic violence included economic control, economic exploitation and employment sabotage. Among those questions, " Has your spouse or any of the people you have been with prevented you from working or forced you to leave a job against your will?" is related to employment sabotage; " Have they ever refused to give you money for you to fulfil the needs of the household even though they have enough money for some other expenses?" is related to economic control and "Have they ever tried to take your own money (in a situation where you have an income) against your own will?" is related to economic exploitation [24].

The state of being exposed to economic violence measured by these questions was used to create the dependent variable. Economic violence was measured with two dichotomous variables. Exposure to economic violence was coded as yes if a woman has experienced at least one of the three substances, and no if a woman has experienced none of the three substances (1 = yes, 0 = no).

## Independent variables

The independent variables in the study were determined by conducting research in the literature. The variables related to sociodemographic and economic characteristics of women were the year of the surveys (2008, 2014), the region (west, south, middle, north, east), the woman's place of residence (rural, urban), age (15–24, 25–34, 35–44, 45–54, 55 and over), educational level (illiterate, elementary school, secondary school, high school, university), employment status (unemployed, employed), relationship status (have a relationship, have no existing relationship, still married), health status (excellent/good, reasonable, bad/very bad), number of children (has no child, one child, two or more), and the status of exposure to violence by first degree relatives (no, yes).

The factors related to women's husbands/partners were husband/partner's educational level (illiterate, elementary school, secondary school, high school, university), the husband/partner's employment status (unemployed, employed), the husband/partner's alcohol use status (no, yes), the husband/partner's gambling status (no, yes), the husband/partner's drug use status (no, yes), the status of being cheated on by husband/partner (no, yes), status of exposure to the husband/partner's verbal violence (no, yes), status of exposure to the husband/partner's physical violence (no, yes), and status of exposure to the husband/partner's sexual violence (no, yes).

Ordinal and nominal variables were defined as dummy variables in order to observe the effects of the categories of all variables to be included in binary logistic regression models [66].

## Research method

The survey statistics in Stata 15 (Stata Corporation) were employed to account for the complex sampling design and weights. Weighted analysis was performed. Firstly, the frequency and

percentages were obtained according to the status of exposure to economic violence of women participating in the research by their husbands/partners. Bivariate analyses were also performed to identify relationships between the outcome variable (exposure to economic violence) and various factors. We estimated bivariate relationships by evaluating significant differences using Pearson chi-square tests for categorical variables. Pearson chi-square ($\chi^2$) not only provides information about the importance of observed differences but also about which categories any differences found arise from [67].

Then, the factors that were effective on women's exposure to economic violence were determined using binary logistic regression analysis.

### Ethics approval and consent to participate

We used secondary data for this study. In order to use the micro dataset from the National Research on Domestic Violence against Women in Turkey, the official permission was obtained from the Turkish Statistical Institute. In addition, a "Letter of Undertaking" was given to the Turkish Statistical Institute for the use of the data subjected to the study.

## Results

### Descriptive statistics and chi-square test

It was determined that 27.2% of the women participating in the research have been subjected to at least one of the types of economic violence throughout their lives. The results of sociodemographic and economic factors affecting economic violence against women are presented in Table 1. 28.8% of women participating in the research lived in the western region. 72.3% of women resided in an urban area. The 25–34 age group with the highest participation was followed by the 35–44 age group by 26.7%. It appeared that 49% of women were elementary school graduates, and 78.8% of them were unemployed. It also may be seen in Table 1 that only 9.2% of women were university graduates. It appeared that the health status of 43.3% of women was good, 87.2% of them were still married, and 70.7% of women had two or more children.

According to the chi-square independence test results that may be seen in Table 1, it was determined that there was a significant relationship between economic violence against women and sociodemographic and economic factors.

When the results of the husband/partner's characteristics affecting economic violence against women given in Table 2 were examined, it appeared that husbands/partners of 42.6% of women were elementary school graduates. The husbands/partners of 82.0% of women were employed. Furthermore while 20.8% of husbands/partners drank alcohol, and 2.1% of them gambled. Moreover, the drug use rate of the husbands/partners was observed to be 0.4%. While the ratio of women who were exposed to husband/partner's verbal violence was 43.6%, the ratio of those exposed to husband/partner's physical violence was 37.1%. Furthermore, 14.3% of women were exposed to husband/partner's sexual violence.

According to chi-square independence test results seen in Table 2, it was determined that there was a significant relationship between economic violence against women and the factors related to the husband/partner (excluding husband/partner's employment status).

### Estimation of model

Binary logistic regression analysis was performed to determine the factors affecting economic violence against women. The coefficient values, standard error, P values, and odds ratio (OR) for binary logistic regression analysis results are presented in Tables 3 and 4.

**Table 1. Findings related to sociodemographic and economic factors affecting economic violence against women.**

| Variables | | Exposure to economic violence | | n (%) | χ2 | P |
|---|---|---|---|---|---|---|
| | | **No** | **Yes** | | | |
| **Region** | West | 3786 (28.7) | 1465 (29.0) | 5251 (28.8) | 14.656 | 0.005 |
| | South | 1125 (8.5) | 470 (9.3) | 1595 (8.8) | | |
| | Central | 2939 (22.3) | 1173 (23.3) | 4112 (22.6) | | |
| | North | 1786 (13.5) | 586 (11.6) | 2372 (13.0) | | |
| | East | 3545 (26.9) | 1350 (26.8) | 4895 (26.9) | | |
| **Place of residence** | Rural | 3982(30.2) | 1074(21.3) | 5056 (27.7) | 144.717 | 0.000 |
| | Urban | 9199 (69.8) | 3970 (78.7) | 13169(72.3) | | |
| **Age** | 15–24 | 1857 (14.1) | 711 (14.1) | 2568 (14.1) | 23.915 | 0.000 |
| | 25–34 | 4152 (31.5) | 1676(33.2) | 5828 (32.0) | | |
| | 35–44 | 3468(26.3) | 1405 (27.9) | 4873 (26.7) | | |
| | 45–54 | 2682 (20.3) | 938 (18.6) | 3620 (19.9) | | |
| | 55+ | 1022 (7.8) | 314(6.2) | 1336 (7.3) | | |
| **Educational level** | Illiterate | 2247 (17.0) | 729 (14.5) | 2976 (16.3) | 174.852 | 0.000 |
| | Elementary school | 6303 (47.8) | 2629(52.2) | 8932(49.0) | | |
| | Secondary school | 1210(9.2) | 573 (11.4) | 1783 (9.8) | | |
| | High school | 2004 (15.2) | 848 (16.8) | 2852 (15.7) | | |
| | University | 1417(10.8) | 262(5.2) | 1679 (9.2) | | |
| **Employment status** | Unemployed | 10230(77.6) | 4139(82.1) | 14369(78.8) | 43.232 | 0.000 |
| | Employed | 2951 (22.4) | 905 (17.9) | 3856 (21.2) | | |
| **Relationship status** | Have no existing relationship | 1033(7.8) | 588 (11.7) | 1621(8.9) | 75.225 | 0.000 |
| | Have a relationship | 563(4.3) | 157(3.1) | 720 (4.0) | | |
| | Still married | 11585(87.9) | 4299(85.2) | 15884 (87.2) | | |
| **Health status** | Excellent/Good | 6126(46.5) | 1756 (34.8) | 7882(43.3) | 227.644 | 0.000 |
| | Reasonable | 5309(40.3) | 2326(46.1) | 7635 (41.9) | | |
| | Bad/Very bad | 1742(13.2) | 960 (19.0) | 2702(14.8) | | |
| **Number of children** | Has no child | 1917(14.5) | 533 (10.6) | 2450(13.4) | 50.054 | 0.000 |
| | One child | 2080(15.8) | 812(16.1) | 2892 (15.9) | | |
| | Two and more | 9184 (69.7) | 3699 (73.3) | 12883(70.7) | | |
| **Status of exposure to violence by first degree relatives** | No | 11915(90.4) | 4217(83.6) | 16132(88.5) | 164.742 | 0.000 |
| | Yes | 1264(9.6) | 825(16.4) | 2089(11.5) | | |

Whether there was multicollinearity between independent variables in the model was also tested. It was considered that those with a variance inflation factor (VIF) value of five and above led to a moderate multicollinearity and those with a value of 10 and above led to a high multicollinearity [68]. According to the VIF results presented in Tables 3 and 4, there was no variable that led to multicollinearity problem between the variables.

When Table 3 was examined, the variables of year of survey, region (west, south, central), place of residence, age (15–24, 25–34, 35–44), educational level (elementary school, secondary school, high school), employment status, relationship status (have a relationship, still married), health status (reasonable, bad/very bad), number of children (one child, two or more), and the status of exposure to violence by first degree relatives appeared to be significant.

According to the binary logistic regression analysis, when it was OR<1, the questioned factor (compared to the reference) had little effect on the case investigated. When it was OR >1, it had an increasing effect compared to the reference group [69].

When Table 4 was examined, the variables of husband/partner's educational level (illiterate, elementary school, secondary school, and high school), husband/partner's employment status,

**Table 2. Findings related to the husband/partner's characteristics affecting economic violence against women.**

| Variables | | Exposure to economic violence | | n (%) | χ2 | P |
|---|---|---|---|---|---|---|
| | | No | Yes | | | |
| Husband/partner's educational level | Illiterate | 535(4.1) | 188(3.7) | 723(4.0) | 113.767 | 0.000 |
| | Elementary school | 5484(41.6) | 2273(45.1) | 7757(42.6) | | |
| | Secondary school | 1815(13.8) | 839(16.6) | 2654 (14.6) | | |
| | High school | 3154 (24.0) | 1201(23.8) | 4355(23.9) | | |
| | University | 2180(16.6) | 542(10.7) | 2722(14.9) | | |
| Husband/partner's employment status | No | 2387(18.1) | 890(17.6) | 3277(18.0) | 0.576 | 0.448 |
| | Yes | 10781(81.9) | 4154(82.4) | 14935 (82.0) | | |
| Husband/partner's alcohol use status | No | 10712(81.3) | 3717(73.7) | 14429(79.2) | 127.86 | 0.000 |
| | Yes | 2463(18.7) | 1326(26.3) | 3789 (20.8) | | |
| Husband/partner's gambling status | No | 13029(98.9) | 4804(95.3) | 17833(97.9) | 238.666 | 0.000 |
| | Yes | 142(1.1) | 239(4.7) | 381 (2.1) | | |
| Husband/partner's drug use status | No | 13132(99.7) | 4992(99.1) | 18124(99.6) | 36.998 | 0.000 |
| | Yes | 33(0.3) | 46(0.9) | 79 (0.4) | | |
| Husband/partner's cheating status | No | 12362(93.9) | 4216(83.6) | 16578(91.1) | 473.608 | 0.000 |
| | Yes | 801(4.4) | 825(4.5) | 1626 (8.9) | | |
| Status of exposure to husband/partner's verbal violence | No | 8515(64.6) | 1769(35.1) | 10284(56.4) | 1293.769 | 0.000 |
| | Yes | 4666(35.4) | 3275(64.9) | 7941 (43.6) | | |
| Status of exposure to husband/partner's physical violence | No | 9284(64.6) | 2181(35.1) | 11465(56.4) | 1156.26 | 0.000 |
| | Yes | 3897(29.6) | 2863(56.8) | 6760 (37.1) | | |
| Status of exposure to husband/partner's sexual violence | No | 11966(90.8) | 3646(72.3) | 15612(85.7) | 1016.103 | 0.000 |
| | Yes | 1211(9.2) | 1395(27.7) | 2606(14.3) | | |

husband/partner's alcohol use, gambling and drug use status, husband/partner's cheating status, and the status of exposure to husband/partner's verbal, physical and sexual violence appeared to be significant.

## Marginal effects

Marginal effects and standard errors of sociodemographic and economic factors affecting economic violence against women are presented in Table 5.

According to Table 5, it appears that the women who participated in the study in 2014 had a higher possibility of exposure to economic violence by 9.95% compared to the participants in 2008. The possibility of exposure to economic violence of women living in the western, southern, and central regions was higher by 15.12%, 15.96%, and 7.98%, respectively, compared to those living in the eastern region. Similarly, women living in urban had a higher possibility of exposure to economic violence by 44.76% comparing to others.

The fact that women were in the 15–24 age group increased the possibility of exposure to economic violence by 54.56% compared to the reference group. The possibility of exposure to economic violence decreased as age increased. Women who graduated from elementary school, secondary and high school had a higher possibility of exposure to economic violence by 18.34%, 25.14% and 19.4%, respectively, compared to women who had never finished school. Employed women had a smaller possibility of exposure to economic violence by 24.03% compared to unemployed women.

Women with one child, and two or more children had a higher possibility of exposure to economic violence by 24.45% and 23.19%, respectively. Women who were exposed to violence

**Table 3. Estimated binary logit model results regarding sociodemographic and economic factors affecting women's exposure to economic violence.**

| Variables | | VIF | β | Std. Error | P | OR |
|---|---|---|---|---|---|---|
| **Constant** | | - | 2.984 | 0.162 | 0.000 | 0.051 |
| **Survey year (reference category: 2008)** | | | | | | |
| | 2014 | 1.04 | 0.138 | 0.046 | 0.003 | 1.148 |
| **Region (reference category: east)** | | | | | | |
| | West | 1.78 | 0.209 | 0.063 | 0.001 | 1.232 |
| | South | 1.30 | 0.221 | 0.077 | 0.004 | 1.247 |
| | Central | 1.58 | 0.109 | 0.064 | 0.087 | 1.115 |
| | North | 1.43 | 0.115 | 0.076 | 0.127 | 1.122 |
| **Place of residence (reference category: rural)** | | | | | | |
| | Urban | 1.12 | 0.602 | 0.052 | 0.000 | 1.825 |
| **Age (reference category: 55+)** | | | | | | |
| | 15–24 | 3.49 | 0.759 | 0.122 | 0.000 | 2.137 |
| | 25–34 | 4.30 | 0.45 | 0.102 | 0.000 | 1.567 |
| | 35–44 | 3.73 | 0.437 | 0.099 | 0.000 | 1.548 |
| | 45–54 | 3.04 | 0.135 | 0.100 | 0.176 | 1.145 |
| **Educational level (reference category: illiterate)** | | | | | | |
| | Elementary school | 2.50 | 0.252 | 0.075 | 0.001 | 1.286 |
| | Secondary school | 1.83 | 0.349 | 0.104 | 0.001 | 1.418 |
| | High school | 2.53 | 0.267 | 0.101 | 0.008 | 1.306 |
| | University | 2.63 | -0.14 | 0.138 | 0.300 | 0.867 |
| **Employment status (reference category: unemployed)** | | | | | | |
| | Employed | 1.14 | -0.33 | 0.059 | 0.000 | 0.720 |
| **Relationship status (reference category: have no existing relationship)** | | | | | | |
| | Have a relationship | 1.62 | -0.29 | 0.146 | 0.048 | 0.750 |
| | Still married | 1.70 | -0.33 | 0.083 | 0.000 | 0.717 |
| **Health status (reference category: excellent/good)** | | | | | | |
| | Reasonable | 1.25 | 0.237 | 0.051 | 0.000 | 1.267 |
| | Bad/Very bad | 1.37 | 0.342 | 0.071 | 0.000 | 1.408 |
| **Number of children (reference category: has no child)** | | | | | | |
| | One child | 2.45 | 0.333 | 0.102 | 0.001 | 1.395 |
| | Two and more | 3.41 | 0.315 | 0.100 | 0.002 | 1.371 |
| **Status of exposure to violence by first degree relatives (reference category: no)** | | | | | | |
| | Yes | 1.04 | 0.249 | 0.069 | 0.000 | 1.282 |

VIF: Variance Inflation Factor; Std. Error: Standard Error; OR: Odds Ratio.

by their first-degree relatives had a higher possibility of exposure to economic violence by 17.41% compared to others.

Marginal effects and standard errors of the factors related to husband/partner affecting economic violence against women are presented in Table 6.

Women whose husbands/partners were illiterate a possibility of exposure to economic violence by 19.91% compared to the reference group. Women whose husbands/partners were high school graduates and university graduates had a higher possibility of exposure to economic violence by 11.66% and 14.57%, respectively, compared to the reference group.

Women whose husbands/partners used alcohol had higher possibility of exposure to economic violence by 7.2% compared to others. Similarly, women whose husbands/partners gambled had a higher possibility of exposure to economic violence by 53.35% compared to others.

**Table 4. Estimated binary logit model results of the factors related to husband/partner affecting women's exposure to economic violence.**

| Variables | | VIF | β | Std. Error | P | OR |
|---|---|---|---|---|---|---|
| **Husband/partner's educational level (reference category: elementary school)** | | | | | | |
| | Illiterate | 1.16 | -0.274 | 0.130 | 0.035 | 0.760 |
| | Secondary school | 1.22 | -0.020 | 0.068 | 0.764 | 0.980 |
| | High school | 1.49 | -0.163 | 0.064 | 0.012 | 0.850 |
| | University | 1.90 | -0.202 | 0.091 | 0.025 | 0.817 |
| **Husband/partner's employment status (reference category: no)** | | | | | | |
| | Yes | 1.12 | 0.032 | 0.062 | 0.605 | 1.032 |
| **Husband/partner's alcohol use status (r reference category: no)** | | | | | | |
| | Yes | 1.17 | 0.101 | 0.056 | 0.073 | 1.106 |
| **Husband/partner's gambling status (reference category: no)** | | | | | | |
| | Evet | 1.07 | 0.831 | 0.149 | 0.000 | 2.296 |
| **Husband/partner's drug use status (reference category: no)** | | | | | | |
| | Yes | 1.03 | 0.401 | 0.291 | 0.168 | 1.493 |
| **Husband/partner's cheating status (reference category: no)** | | | | | | |
| | Yes | 1.15 | 0.382 | 0.073 | 0.000 | 1.465 |
| **Status of exposure to husband/partner's verbal violence (reference category: no)** | | | | | | |
| | Yes | 1.45 | 0.634 | 0.054 | 0.000 | 1.885 |
| **Status of exposure to husband/partner's physical violence (reference category: no)** | | | | | | |
| | Yes | 1.56 | 0.581 | 0.056 | 0.000 | 1.787 |
| **Status of exposure to husband/partner's sexual violence (reference category: no)** | | | | | | |
| | Yes | 1.26 | 0.742 | 0.063 | 0.000 | 2.099 |
| **Classification success** | | | | | | 0.754 |
| **Pseudo R$^2$** | | | | | | 0.121 |
| **Cox-Snell/ML R$^2$** | | | | | | 0.133 |
| **AIC** | | | | | | 18891.032 |
| **BIC** | | | | | | 19164.257 |
| **Log-likelihood** | | | | | | -9410.516 |
| **N** | | | | | | 18150 |

VIF: Variance Inflation Factor; Std. Error: Standard Error; OR: Odds Ratio.

Women who were cheated on by their husbands/partners had a higher possibility of exposure to economic violence by 26.29% compared to others. Women who were exposed to verbal, physical and sexual violence by their husbands/partners had higher possibility of exposure to economic violence by 45.35%, 41.07%, and 49.56%, respectively, compared to others.

## Discussion

To make women economically dependent on men causes them to be significantly exposed to violence, physically, psychologically, sexually, and economically [70]. In some studies, it was determined that most of the husbands used violence against women and that women had health problems. It was determined that the vast majority of women were depressed or nervous [71]. Therefore, many forms of violence faced by women are based on economic violence [72]. In with this study, sociodemographic, economic, and husband/partner-related factors affecting economic violence against women in Turkey were detected.

The study determined that the educational status of women was effective for being exposed to economic violence. The study concluded that as the level of education increased, the possibility of being exposed to economic violence did not decrease. Increase in the level of

**Table 5. Marginal effects for sociodemographic and economic factors affecting women's exposure to economic violence.**

| Variables | Marginal effects (%) | Std. Error |
|---|---:|---:|
| **Survey year (reference category: 2008)** | | |
| 2014 | 9.85[a] | 0.033 |
| **Region (reference category: east)** | | |
| West | 15.12[a] | 0.046 |
| South | 15.96[a] | 0.055 |
| Central | 7.98[b] | 0.047 |
| North | 8.46 | 0.055 |
| **Place of residence (reference category: rural)** | | |
| Kent | 44.76[a] | 0.040 |
| **Age (reference category: 55+)** | | |
| 15–24 | 54.56[a] | 0.089 |
| 25–34 | 33.48[a] | 0.078 |
| 35–44 | 32.59[a] | 0.076 |
| 45–54 | 10.42 | 0.077 |
| **Educational level (reference category: illiterate)** | | |
| Elementary school | 18.34[a] | 0.056 |
| Secondary school | 25.14[a] | 0.075 |
| High school | 19.4[a] | 0.074 |
| University | -10.87 | 0.105 |
| **Employment status (reference category: unemployed)** | | |
| Employed | -24.03[a] | 0.044 |
| **Relationship status (reference category: have no existing relationship)** | | |
| Have a relationship | 19.83[b] | 0.103 |
| Still married | 23.06[a] | 0.056 |
| **Health status (reference category: excellent/good)** | | |
| Reasonable | 17.09[a] | 0.037 |
| Bad/Very bad | 24.37[a] | 0.050 |
| **Number of children (reference category: has no child)** | | |
| One child | 24.45[a] | 0.076 |
| Two and more | 23.19[a] | 0.075 |
| **Status of exposure to violence by first degree relatives (reference category: no)** | | |
| Evet | 17.41[a] | 0.041 |

[a]$p < .01$

[b]$p < .05$; Std. Error: Standard Error.

education was not completely a preventative factor of economic violence against women [73]. There are studies in the literature in which similar results have been obtained. It has been found that economic violence is more common in women with higher education [25, 74].

In the study, it was determined that the husband/partner's educational status had an effect on women's exposure to economic violence. In the study, it was concluded that the possibility of exposure to economic violence decreased as women's husbands/partners' educational level increased. In a study conducted in Ghana, it was reported that a high educational level of the husbands/partners had a lessening effect on violence [75].

Women's employment status also affects the economic violence. It was determined that unemployed women were more exposed to economic violence compared to employed

**Table 6. Marginal effects for the factors related to husband/partner affecting women's exposure to economic violence.**

| Variables | | Marginal effects (%) | Std. Error |
|---|---|---|---|
| **Husband/partner's educational level (reference category: elementary school)** | | | |
| | Illiterate | 19.91[b] | 0.097 |
| | Secondary school | -1.43 | 0.048 |
| | High school | 11.66[b] | 0.046 |
| | University | 14.57[b] | 0.066 |
| **Husband/partner's employment status (reference category: no)** | | | |
| | Yes | 2.29 | 0.044 |
| **Husband/partner's alcohol use status (r reference category: no)** | | | |
| | Yes | 7.20[c] | 0.040 |
| **Husband/partner's gambling status (reference category: no)** | | | |
| | Yes | 53.35[a] | 0.084 |
| **Husband/partner's drug use status (reference category: no)** | | | |
| | Yes | 27.27 | 0.187 |
| **Husband/partner's cheating status (reference category: no)** | | | |
| | Yes | 26.29[a] | 0.048 |
| **Status of exposure to husband/partner's verbal violence (reference category: no)** | | | |
| | Yes | 45.35[a] | 0.038 |
| **Status of exposure to husband/partner's physical violence (reference category: no)** | | | |
| | Yes | 41.07[a] | 0.039 |
| **Status of exposure to husband/partner's sexual violence (reference category: no)** | | | |
| | Yes | 49.56[a] | 0.039 |

[a] $p < .01$
[b] $p < .05$
[c] $p < .10$; Std. Error: Standard Error.

women. Although it was determined that the violence experienced by women when they were actively working was less and that employed women were less exposed to violence compared to unemployed women, it was demonstrated that this factor was not a factor preventing violence against women [76]. The fact that the woman who has a relationship with violence has economic solvency may put her at greater risk of additional violence because economic solvency may threaten the role of male domination [77, 78]. Therefore, women's economic solvency should be supported by important steps such as financial education classes [79]. Furthermore, women's economic solvency can be a factor that prevents violence and protects women if it is supported by more comprehensive programs such as government support [80]. Poor women are exploited more than women with a high income. However, the causes of violence against women contain more components than only the economic dimension [81].

In the study, it was determined that the place of residence of women was among the determinants of economic violence and that the region of residence of women had an effect on exposure to economic violence. Women residing in urban areas had a higher possibility of exposure to economic violence compared to those living in rural. Furthermore, the region of residence of women also affected the economic violence. It was detected that the women living in the western, central and southern regions had a higher possibility of exposure to economic violence compared to women living in the eastern region.

It was determined that the number of children was also effective on women's exposure to economic violence. Women with children had a higher possibility of exposure to violence

compared to women with no children. However, it was determined that the possibility of exposure to violence was slightly decreased as the number of children increased. In some studies, it was also observed that the women with fewer children were more exposed to economic violence compared to those with many children, along with the increase in women's employment status [82, 83].

One of the remarkable results of the study was that the variable of marital status was effective on women's exposure to economic violence. It was determined that the women who were still married or had a relationship had higher possibility of exposure to economic violence by their husbands/partners compared to women who had no existing relationship. One of the variables affecting women's exposure to economic violence was health status. It was concluded that the possibility of exposure to economic violence of women decreased as their health status improved.

In the study, it was determined that the women whose husbands/partners used alcohol were more exposed to economic violence. According to the results of some studies, alcohol consumption does not automatically lead to violence. In other words, alcohol consumption is not considered a primary factor causing violence but as a situational factor that intensifies the conflict between the couples [84, 85]. In many studies, it was also determined that there was a relationship between alcohol consumption and the violent behavior of husbands/partners [86–88].

In the studies on the factors affecting economic violence against women, it was determined that situations such as a childhood experience of violence, witnessing violence at a young age, and poor mental health were effective for women's exposure to economic violence [29]. Individuals who have experienced or witnessed violence in childhood appear to be individuals who are prone to violence against their wives in the future [89].

In the study, it was determined that the women whose husbands/partners were drug users were more exposed to economic violence compared to the women whose husbands/partners did not use drugs. It was determined that the women whose husbands/partners were gamblers were more exposed to economic violence compared to the women whose husbands/partners who did not gamble. Similar results were also achieved in some studies [90, 91].

It was determined that the women who were exposed to violence by their first-degree relatives were more exposed to economic violence compared to those who were not exposed. Furthermore, it was determined that the women who were cheated on by their husbands/partners had a higher possibility of exposure to economic violence compared to those who were not cheated on.

Other forms of violence were also effective for women's exposure to economic violence. It was determined that the women who were exposed to verbal, sexual, and physical violence by their husbands/partners had a higher possibility of exposure to economic violence. It is important to understand that there are different types of partner violence against women and that there is a cause-and-effect relationship between them. Environments that feed and witness violence will increase violent behavior, and these effects will determine the direction of anti-violence efforts.

The results obtained in this study are important in that they can be a source of information for establishing policies and programs to prevent violence against women. This study can also be a significant guide in determining the priority areas for the resolution of economic violence against women.

With the policies to prevent economic violence against women in Turkey, more effective results can be obtained by giving priority to groups such as, women living in regions with higher levels of wealth, the 15–24 age group, primary school graduates, those with poor/very poor health, those with children, those who have been exposed to violence from first-degree

relatives, those who have a partner using alcohol/drugs and gambling, and those who have been exposed to sexual violence.

For the resolution of violence against women, social support should be mentioned again as a means of meeting women's economic obligations [82, 83, 92, 93]. Networking is required to help in finding a job [93], and social support is required to provide childcare [82].

Development initiatives to improve the access of girls and boys to education may play a significant role in the prevention of violence [94]. To raise the awareness of children at a young age of the subjects that express what women mean for the society by means of including it, in the students' course books, or if necessary, with the course books entitled "Woman" during the academic year in line with the studies indicating that violence is associated with childhood will probably be the most basic solution for prevent violence.

Nowadays, although laws exist than aim to prevent all forms of violence against women, these regulations are inadequate in practice. Public institutions, local administrations, the providers of law, and the organizations such as the media should take responsibility for the prevention of violence against women and take the necessary steps in this regard. It is not enough to design such organizations just to eliminate violence. The steps they take to reduce violence against women should be more stable with more robust decisions. Otherwise, it is obvious that the studies that have been conducted to prevent violence against women for many years will have no effect.

It is necessary to support individuals who will say that we must stop this violence that the whole world feels to the bone, especially all kinds of institutions, organizations, and policies that may reflect women's voices. An overall struggle may lead to healthy and consistent results. Thus, women, in particular, need to create the necessary environment to consider themselves individuals and without the pressures from traditional society. Although the existence of a world where women can live as men requires a long process, it is necessary to struggle for it. Mutual dependence is different from interdependence [95] because mutual dependence weakens the woman, while interdependence makes her strong. Unless gender equality and the social structure to support it are formed and there is a change in the mentality of the society, there will be no specific and clear solutions to violence against women.

Domestic violence victims, especially women with low-income, lack the resources needed to start a new life for themselves and their children. Women with sufficient resources will find it easier to live independently and avoid returning to an abusive partner. Ensuring independence may block the abuser's goals of establishing and maintaining control. Interventions must therefore be carefully developed, with security at the center.

An economic empowerment program should be developed that focuses both on increasing basic financial knowledge and skills and on empowering victims by increasing victims' confidence in their ability to manage their own finances and to develop security plans for their financial future.

The results of the study provide various recommendations for policymakers. Economic violence is predictable and preventable. Ending economic violence requires the use of long-term and multiple strategies, involving all segments of society. Policymakers should understand the consequences of economic violence and establish policies that support survivors and prohibit economic violence. It is believed that the results of the study will be a guide for local governments, official and voluntary organizations, educational institutions and relevant researchers in the prevention of economic violence against women.

Research is needed to understand the characteristics of individuals who perpetrate economic violence against their partners and to identify risk and protective factors around economic violence behavior. More research is needed to address the effects of economic violence and prevent economic violence in order to improve women's safety.

This study has several limitations. Firstly, the data in the study were secondary data. The variables required for statistical analysis consisted of the variables in the dataset. The variables such as income level, marriage age of woman and her husband/partner could not be included in the model. Secondly, since the data are cross-sectional, it was not possible to reveal a definitive causal relationship related to the factors affecting exposure to economic violence. The third one is that the data obtained in the study were women's own answers. Since there is no officially recorded data, the results obtained in the data collection method may be biased. Finally, the question paper for the study of the National Research on Domestic Violence against Women in Turkey was designed by taking into account the question papers used by the World Health Organization's Multi-country Study on Women's Health and Domestic Violence Against Women Study. Therefore, only three questions were asked to women about economic violence.

## Acknowledgments

The authors would like to thank the Turkey Statistical Institute for the data. The views and opinions expressed in this manuscript are those of the authors only and do not necessarily represent the views, official policy, or position of the Turkey Statistical Institute.

## Author Contributions

**Conceptualization:** Ömer Alkan.

**Data curation:** Ömer Alkan.

**Formal analysis:** Ömer Alkan.

**Investigation:** Şeyda Ünver.

**Methodology:** Ömer Alkan, Şenay Özar.

**Resources:** Şenay Özar, Şeyda Ünver.

**Writing – original draft:** Şenay Özar.

**Writing – review & editing:** Ömer Alkan, Şenay Özar, Şeyda Ünver.

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
