## [Decision Letter · Decision Letter 0]

2 Aug 2020

PONE-D-20-15042

Economic Violence against Women: A case in Turkey

PLOS ONE

Dear Dr. Alkan,

Thank you for submitting your manuscript to PLOS ONE. After careful consideration, we feel that it has merit but does not fully meet PLOS ONE’s publication criteria as it currently stands. Therefore, we invite you to submit a revised version of the manuscript that addresses the points raised during the review process.

We look forward to receiving your revised manuscript.

Kind regards,

M Niaz Asadullah

Academic Editor

PLOS ONE

Journal Requirements:

2. Please ensure that your references are formatted according to the PLOS ONE submission guidelines https://journals.plos.org/plosone/s/submission-guidelines#loc-references. In particular, please note that references should be listed at the end of the manuscript and numbered in the order that they appear in the text. In the text, please cite the reference number in square brackets.

3.In your Data Availability statement, you have not specified where the minimal data set underlying the results described in your manuscript can be found. PLOS defines a study's minimal data set as the underlying data used to reach the conclusions drawn in the manuscript and any additional data required to replicate the reported study findings in their entirety. All PLOS journals require that the minimal data set be made fully available. For more information about our data policy, please see http://journals.plos.org/plosone/s/data-availability.

Additional Editor Comments (if provided):

Both referees read the paper with interest but share the view that substantive changes are necessary before the paper can be considered for publication. I have also gone through the paper and share the referee recommendations. Referee 2 has emphasized on the need to engage better with the literature and be more critical in discussing the key concepts & measures (e.g. women’s exposure to economic IPV) and provided a detailed revision guideline. Other recommendations include a thorough language check to correct grammatical mistakes.

I hope all of these addressed during the revision stage. I also have a few other points for the authors to consider.

1. In the discussion section (2nd last para), the authors say: "Women's empowerment, and to consider them as individuals existing in society are important steps to prevent violence. Unless a society that understands that women can have economic potential and take part in successful jobs is formed, violence against women should not be expected to end completely." Pls avoid such unsubstantiated claims and viewpoints. There's already a literature on the link between empowerment and violence. Kindly cite relevant studies in support of your claim.

2. tables 3-4: pls report sample size. also no need to report confidence intervals

3. the paper would also benefit from a section on study background and context so that readers unfamiliar with Turkey has an introduction to norms of violence , gender roles and power balance within marriage in Turkey.

Reviewers' comments:

Reviewer's Responses to Questions

**Comments to the Author**

1. Is the manuscript technically sound, and do the data support the conclusions?

Reviewer #1: Partly

Reviewer #2: Partly

2. Has the statistical analysis been performed appropriately and rigorously? 

Reviewer #1: No

Reviewer #2: No

3. Have the authors made all data underlying the findings in their manuscript fully available?

Reviewer #1: No

Reviewer #2: Yes

4. Is the manuscript presented in an intelligible fashion and written in standard English?

Reviewer #1: No

Reviewer #2: Yes

5. Review Comments to the Author

Reviewer #1: The subject of manuscripts appears interesting, but some important revisions for clarity are needed before publication decision.

The hypothesis of the research are not clear. The author(s) needs to explain what are the main research questions of this paper. This paper needs a comprehensive literature review on the issue of conceptualization of economic violence. The hypothesis address this issue and the main contribution to literature should be explained.

It is sufficient to present ratios for descriptive statistics. It is not clear why the author(s) apply Chi-square tests such as the Chi-square test between Survey year and “Exposure to Economic Violence”.

The paper use “The binary logistic regression” methodology. It present “odd ratios”. But, it could be better if the author(s) calculates “marginal effects” and add them to the manuscript.

The manuscript includes unnecessary explanations about Tables. For example Table 3 includes odds ratios and confidence intervals. However, the following paragraphs repeat all these statistics. It will be better to explain the summary of the results and their connection with the hypothesis. The dependent variable is created with respect to answers of the women about their husband/partner’s behavior. So, it seems that the research includes women who live with their husbands/partners. Some of the explanations lose its meaning such as “The women who had a relationship and were married had a higher odds ratio of exposure to economic violence compared to women who had no existing relationship.” On the other hand, the data source of the research is limited. The subject in this research needs to collect data using interviews with women. For example Sen, S. and Bolsoy, N (2017) collect data using questionnaires and “Scale of Domestic Violence against Women”.

There are some contradictions in the manuscript. For example, in the abstract, the explanation about education level as follow: “Women who graduated from elementary school, secondary school, and high school had a higher odds ratio of exposure to economic violence compared to those who have never gone to school.” P.19 “Women who graduated from elementary school, secondary and high school had higher possibility of exposure to economic violence by 18.34%, 25.14% and 19.4%, respectively, compared to women who have never finished school.” However P.21 there is the following explanation: “In the study, it was determined that the educational status of women had an effect on exposure to economic violence. In the study, it was concluded that the possibility of exposure to economic violence decreased as the educational level increased.”

The author(s) claims that the paper can be a significant guide in determining the priority areas for the solution of economic violence. However, the connections between the results of research and priority area explanations in the conclusion are weak. It needs to explain relations between the results and priority area explanations.

The authors also needs to read the text carefully and should correct grammatical mistakes. It needs to proof reading before to submitting.

Reviewer #2: Comments on

This study examined the correlates of women’s exposure to economic coercion in Turkey using the micro data set of the National Research on Domestic Violence against Women in Turkey. The topic of women’s exposure to economic coercion is salient and understudied, so this manuscript makes an important empirical contribution that is worthy of publication and that will be of interest to researchers of violence against women and women’s health. Below are suggestions that the authors might consider to improve the contributions that this manuscript makes.

Introduction/Background

The literature review should be strengthened considerably by including some of the seminal conceptual and empirical contributions to the field. For example, the authors may consider citing the following studies (there are many others, so these are illustrative):

Conceptualization and reviews of economic coercion/IPV. Here, the authors should offer a clear definition of economic IPV/coercion. What are the dimensions of economic IPV, as discussed in the literature? Providing a clear definition will allow the authors to critique how well economic IPV is measured in their data source (see comment about the discussion).

1. Dutton, M. A., & Goodman, L. A. (2005). Coercion in intimate partner violence: Toward a new conceptualization. Sex roles, 52(11-12), 743-756.

2. Postmus, J. L., Plummer, S. B., McMahon, S., Murshid, N. S., & Kim, M. S. (2012). Understanding economic abuse in the lives of survivors. Journal of interpersonal violence, 27(3), 411-430.

3. Postmus, J. L., Hoge, G. L., Breckenridge, J., Sharp-Jeffs, N., & Chung, D. (2020). Economic abuse as an invisible form of domestic violence: A multicountry review. Trauma, Violence, & Abuse, 21(2), 261-283.

4. Stylianou, A. M. (2018). Economic abuse within intimate partner violence: A review of the literature. Violence and Victims, 33(1), 3-22.

Measurement of economic coercion/IPV:

5. Postmus, J. L., Plummer, S. B., & Stylianou, A. M. (2016). Measuring economic abuse in the lives of survivors: Revising the Scale of Economic Abuse. Violence against women, 22(6), 692-703.

6. Adams, A. E., Beeble, M. L., & Gregory, K. A. (2015). Evidence of the construct validity of the Scale of Economic Abuse. Violence and victims, 30(3), 363-376.

7. Adams, A. E., Greeson, M. R., Littwin, A. K., & Javorka, M. (2019). The Revised Scale of Economic Abuse (SEA2): Development and initial psychometric testing of an updated measure of economic abuse in intimate relationships. Psychology of Violence.

8. Schrag, R. J. V., & Ravi, K. (2020). Measurement of economic abuse among women not seeking social or support services and dwelling in the community. Violence and victims, 35(1), 3-19.

Correlates of economic IPV and co-occurrence with other forms of IPV

9. Yount, K. M., Krause, K. H., & VanderEnde, K. E. (2016). Economic coercion and partner violence against wives in Vietnam: a unified framework?. Journal of interpersonal violence, 31(20), 3307-3331.

Health effects of economic IPV against women:

10. Khan, Z., Cheong, Y. F., Miedema, S., Naved, R. T., & Yount, K. (2020). Women's Experiences of Intimate Partner Violence, Economic Coercion, and Depressive Symptoms in Bangladesh. Economic Coercion, and Depressive Symptoms in Bangladesh (January 31, 2020).

11. Adams, A. E., & Beeble, M. L. (2019). Intimate partner violence and psychological well-being: Examining the effect of economic abuse on women’s quality of life. Psychology of violence, 9(5), 517.

12. Davila, A. L., Johnson, L., & Postmus, J. L. (2017). Examining the relationship between economic abuse and mental health among Latina intimate partner violence survivors in the United States. Journal of interpersonal violence, 0886260517731311.

13. Stylianou, A. M. (2018). Economic abuse experiences and depressive symptoms among victims of intimate partner violence. Journal of family violence, 33(6), 381-392.

14. Voth Schrag, R. J., Robinson, S. R., & Ravi, K. (2019). Understanding pathways within intimate partner violence: Economic abuse, economic hardship, and mental health. Journal of Aggression, Maltreatment & Trauma, 28(2), 222-242.

Data/Methods

The authors should provide more detail on the data source. What were the inclusion criteria? Was this a cross-sectional study? What was the sampling strategy? What was the response rate?

The authors also should provide a critical discussion of the three questions on economic IPV? How well do they align with definitions of economic coercion/IPV? What aspects of economic coercion/IPV are captured? What are missed? What was the time frame for which participants were asked to report on experiences of economic IPV—lifetime? Prior year? Why did the authors select a binary outcome instead of a count outcome (number of types of economic coercion experienced in the given time frame?)

The authors say that they relied on the literature for decisions about selections for the correlates of IPV, but do not cite any literature in the discussion of correlates. I refer the authors here to the article by Yount et al. in Vietnam as a relevant reference in this section.

Analytically, I wonder about the extent to which other forms of violence “predict” economic coercion, or whether these are co-occurring and reciprocally influential behaviors in an abusive relationship. The temporal ordering of the variables is critical, but is not well described. I encourage the authors to be more transparent about the timeframe for which different forms of violence are measured, and the temporal ordering of economic coercion with other forms of IPV used as predictors.

Findings/Discussion

The authors should do a better job of critiquing the measure of women’s exposure to economic IPV. What are the implications of poor measurement for inferences from the analysis? What recommendations come out of the limitations of the measure of economic IPV used in the study?

6. PLOS authors have the option to publish the peer review history of their article (what does this mean?). If published, this will include your full peer review and any attached files.

Reviewer #1: No

Reviewer #2: No

---

## [Author Response · Author response to Decision Letter 0]

7 Dec 2020

Additional Editor Comments (if provided):

Both referees read the paper with interest but share the view that substantive changes are necessary before the paper can be considered for publication. I have also gone through the paper and share the referee recommendations. Referee 2 has emphasized on the need to engage better with the literature and be more critical in discussing the key concepts & measures (e.g. women’s exposure to economic IPV) and provided a detailed revision guideline. Other recommendations include a thorough language check to correct grammatical mistakes.

I hope all of these addressed during the revision stage. I also have a few other points for the authors to consider.

1. Comment: In the discussion section (2nd last para), the authors say: "Women's empowerment, and to consider them as individuals existing in society are important steps to prevent violence. Unless a society that understands that women can have economic potential and take part in successful jobs is formed, violence against women should not be expected to end completely." Pls avoid such unsubstantiated claims and viewpoints. There's already a literature on the link between empowerment and violence. Kindly cite relevant studies in support of your claim.

1. Response: In line with the editor's criticism, the relevant paragraph has been removed from the paper.

2. Comment: tables 3-4: pls report sample size. also no need to report confidence intervals

2. Response: Table 3-4 has been reorganized and confidence intervals have been removed. The sample volume is also indicated in the table.

3. Comment: the paper would also benefit from a section on study background and context so that readers unfamiliar with Turkey has an introduction to norms of violence, gender roles and power balance within marriage in Turkey.

3. Response: The necessary correction has been made in accordance with the editor's recommendation. For this purpose, the title “Economic violence/abuse and gender roles in Turkey” was added into the introduction section.

Reviewers' comments:

Reviewer #1: 

The subject of manuscripts appears interesting, but some important revisions for clarity are needed before publication decision.

1. Comment: The hypothesis of the research are not clear. The author(s) needs to explain what are the main research questions of this paper. This paper needs a comprehensive literature review on the issue of conceptualization of economic violence. The hypothesis address this issue and the main contribution to literature should be explained.

1. Response: In accordance with the criticism of the Reviewer, the main research questions of the paper were added to the introduction section. A comprehensive literature research was carried out and the title “Literature review” was added into the paper.

2. Comment: It is sufficient to present ratios for descriptive statistics. It is not clear why the author(s) apply Chi-square tests such as the Chi-square test between Survey year and “Exposure to Economic Violence”.

2. Response: The necessary corrections were made in accordance with the criticism of the Reviewer. The survey year variable is removed from the chi-square analysis Table. Necessary explanations have been made about the chi-square test. In accordance with the criticism, the description “Bivariate analyses were also performed to identify relationships between the outcome variable (exposure to economic violence) and various factors. We estimated bivariate relationships by evaluating significant differences using Pearson chi-square tests for categorical variables. Pearson chi-square (χ2) not only provides information about the importance of observed differences, but also provides detailed information about which categories any differences found arise from” was added into the paper.

3. Comment: The paper use “The binary logistic regression” methodology. It present “odd ratios”. But, it could be better if the author(s) calculates “marginal effects” and add them to the manuscript.

3. Response: In accordance with the Reviewer's criticism, “odd ratios” comments were removed and marginal effects were interpreted in the paper.

4. Comment: The manuscript includes unnecessary explanations about Tables. For example Table 3 includes odds ratios and confidence intervals. However, the following paragraphs repeat all these statistics. It will be better to explain the summary of the results and their connection with the hypothesis. The dependent variable is created with respect to answers of the women about their husband/partner’s behavior. So, it seems that the research includes women who live with their husbands/partners. Some of the explanations lose its meaning such as “The women who had a relationship and were married had a higher odds ratio of exposure to economic violence compared to women who had no existing relationship.” On the other hand, the data source of the research is limited. The subject in this research needs to collect data using interviews with women. For example Sen, S. and Bolsoy, N (2017) collect data using questionnaires and “Scale of Domestic Violence against Women”.

4. Response: Table 3 comments were revised in accordance with the Reviewer's criticism. In accordance with the criticism, the results have been reinterpreted. In the paper, our explanation on the dependent variable was insufficient. “Has your spouse or any of the people you have been with ...” has been added into the paper regarding the dependent variable. Additional explanations were made about the "relationship status" variable in the paper. The phrase “In this study, women who were married, had a relationship or had any previous relationship were included in the analysis. Single women who had never been in a relationship before were not included in the study.” was added to the paper. Secondary data was used in the paper. “The National Research on Domestic Violence against Women in Turkey” data was used with official permission from the Turkish Statistical Institute. “In order to understand and determine the causes the different dimensions of violence against women, and also to fulfill the need to collect data on this subject, for the first time in 2008, a comprehensive "The National Research on Domestic Violence against Women in Turkey " was held. " The National Research on Domestic Violence against Women in Turkey" which was carried out in 2014 is important in terms of reflecting the violence against women during the time change. “The National Research on Domestic Violence against Women in Turkey” is one of the most comprehensive studies conducted nationwide in order to understand the extent, content, causes and consequences of domestic violence experienced by women, as well as the risk factors” explanation has been added into the paper about the reason why this dataset is used.

5. Comment: There are some contradictions in the manuscript. For example, in the abstract, the explanation about education level as follow: “Women who graduated from elementary school, secondary school, and high school had a higher odds ratio of exposure to economic violence compared to those who have never gone to school.” P.19 “Women who graduated from elementary school, secondary and high school had higher possibility of exposure to economic violence by 18.34%, 25.14% and 19.4%, respectively, compared to women who have never finished school.” However P.21 there is the following explanation: “In the study, it was determined that the educational status of women had an effect on exposure to economic violence. In the study, it was concluded that the possibility of exposure to economic violence decreased as the educational level increased.”

5. Response: In accordance with the criticism of the Reviewer, the contradiction in Page 19 and Page 21 in the paper was corrected and written accordingly.

6. Comment: There The author(s) claims that the paper can be a significant guide in determining the priority areas for the solution of economic violence. However, the connections between the results of research and priority area explanations in the conclusion are weak. It needs to explain relations between the results and priority area explanations.

6. Response: The discussion section was reorganized in accordance with the criticism of the Reviewer. The results found are re-discussed in detail.

7. Comment: The authors also needs to read the text carefully and should correct grammatical mistakes. It needs to proof reading before to submitting.

7. Response: Language and grammatical errors were corrected during the “proofreading” made by the editor, whose mother tongue is English, in accordance with the criticism of the Reviewer.

Reviewer #2: 

This study examined the correlates of women’s exposure to economic coercion in Turkey using the micro data set of the National Research on Domestic Violence against Women in Turkey. The topic of women’s exposure to economic coercion is salient and understudied, so this manuscript makes an important empirical contribution that is worthy of publication and that will be of interest to researchers of violence against women and women’s health. Below are suggestions that the authors might consider to improve the contributions that this manuscript makes.

Introduction/Background

1. Comment: The literature review should be strengthened considerably by including some of the seminal conceptual and empirical contributions to the field. For example, the authors may consider citing the following studies (there are many others, so these are illustrative):

1. Response: In accordance with the criticism of the Reviewer, a comprehensive literature research was carried out and the title “Literature review” was added into the paper. The studies recommended by the Reviewer were reviewed in detail and the introduction section was rewritten to be used in the paper.

2. Comment: Conceptualization and reviews of economic coercion/IPV. Here, the authors should offer a clear definition of economic IPV/coercion. What are the dimensions of economic IPV, as discussed in the literature? Providing a clear definition will allow the authors to critique how well economic IPV is measured in their data source (see comment about the discussion).

2. Response: In accordance with the criticism of the Reviewer, the concept of economic violence is explained in detail in the introduction section. The definition of economic violence, its dimensions and how economic violence is measured in the data source are explained in detail. 

3. Comment: The authors should provide more detail on the data source. What were the inclusion criteria? Was this a cross-sectional study? What was the sampling strategy? What was the response rate?

3. Response: Detailed explanations on the data source were written in accordance with the criticism of the Reviewer. Accordingly, phrases “As part of the Violence Survey, Turkey is divided into 30 layers to provide estimations at the level of Country, urban/rural, 12 regions and 5 regions. Except for the Istanbul region, which is one of the 12 regions, approximately 75 percent and 25 percent distribution was made to urban and rural layers. In Istanbul, about 5 percent of households were selected from rural. In the study, settlements with a population of 10,000 and above constitute urban areas, and settlements with a population of less than 10,000 constitute rural areas. The sample of the research is cluster sampling”, “In a 2008 study, 12,795 women were interviewed face-to-face and the female question paper was completed with a rejection rate of 2.1%. The answer rate in female interviews is 86.1%. In the 2014 survey, 7,462 women were interviewed face-to-face and the female question paper was completed, with a rejection rate of 4.4%. The answer rate in female interviews is 83.3%. These data sets included female weights calculated to match the sample design of the study”, and “In this study, women who were married, had a relationship or had any previous relationship were included in the analysis. Single women who had never been in a relationship before were not included in the study. In conclusion, the data of a total of 18,225 women aged 15 years and over who participated in the National Research on Domestic Violence against Women in Turkey, including 11,514 women in 2008 and 6,711 women in 2014, were employed” were added into the paper.

4. Comment: The authors also should provide a critical discussion of the three questions on economic IPV? How well do they align with definitions of economic coercion/IPV? What aspects of economic coercion/IPV are captured? What are missed? What was the time frame for which participants were asked to report on experiences of economic IPV—lifetime? Prior year? Why did the authors select a binary outcome instead of a count outcome (number of types of economic coercion experienced in the given time frame?)

4. Response: The aspects of economic violence have been rewritten in accordance with the criticism of the Reviewer. New information on questions related to economic violence in data source is provided. The time frame for economic violence is written in more detail. Extra explanations are provided related to why the dependent variable is qualitative. 

5. Comment: The authors say that they relied on the literature for decisions about selections for the correlates of IPV, but do not cite any literature in the discussion of correlates. I refer the authors here to the article by Yount et al. in Vietnam as a relevant reference in this section.

5. Response: A comprehensive literature research was conducted in accordance with the criticism of the Reviewer. The study recommended by the Reviewer was examined in detail and used in the paper.

6. Comment: Analytically, I wonder about the extent to which other forms of violence “predict” economic coercion, or whether these are co-occurring and reciprocally influential behaviors in an abusive relationship. The temporal ordering of the variables is critical, but is not well described. I encourage the authors to be more transparent about the timeframe for which different forms of violence are measured, and the temporal ordering of economic coercion with other forms of IPV used as predictors.

6. Response: Detailed explanations on other types of violence have been added in accordance with the criticism of the Reviewer. As a result of literature research, the relationship between economic violence and other types of violence has been extensively discussed. 

7. Comment: The authors should do a better job of critiquing the measure of women’s exposure to economic IPV. What are the implications of poor measurement for inferences from the analysis? What recommendations come out of the limitations of the measure of economic IPV used in the study?

7. Response: In accordance with the criticism of the Reviewer, the discussion section was reorganized and the necessary explanations were added.

---

## [Decision Letter · Decision Letter 1]

3 Mar 2021

Economic Violence against Women: A case in Turkey

PONE-D-20-15042R1

Dear Dr. Alkan,

We’re pleased to inform you that your manuscript has been judged scientifically suitable for publication and will be formally accepted for publication once it meets all outstanding technical requirements.

Kind regards,

Sisi Zhang

Academic Editor

PLOS ONE

Reviewers' comments:

Reviewer's Responses to Questions

**Comments to the Author**

1. If the authors have adequately addressed your comments raised in a previous round of review and you feel that this manuscript is now acceptable for publication, you may indicate that here to bypass the “Comments to the Author” section, enter your conflict of interest statement in the “Confidential to Editor” section, and submit your "Accept" recommendation.

Reviewer #3: All comments have been addressed

Reviewer #4: All comments have been addressed

2. Is the manuscript technically sound, and do the data support the conclusions?

Reviewer #3: Yes

Reviewer #4: Yes

3. Has the statistical analysis been performed appropriately and rigorously? 

Reviewer #3: Yes

Reviewer #4: Yes

4. Have the authors made all data underlying the findings in their manuscript fully available?

Reviewer #3: Yes

Reviewer #4: Yes

5. Is the manuscript presented in an intelligible fashion and written in standard English?

Reviewer #3: Yes

Reviewer #4: Yes

6. Review Comments to the Author

Reviewer #3: I am reviewing authors’ edits to the manuscript and their responses to the previous reviewer #1’s comments. I am happy to see that the authors have addressed all the comments that reviewer #1 has raised and the manuscript is significantly improved. Specifically, the manuscript adds a comprehensive literature review, modifies discussions on “marginal effects” and Pearson chi-square tests, and explains tables better. The contradictory results are corrected, and the discussion section was reorganized.

Reviewer #4: The revised manuscript has addressed all comments raised by previous reviewers. I recommend accepting the paper.

7. PLOS authors have the option to publish the peer review history of their article (what does this mean?). If published, this will include your full peer review and any attached files.

Reviewer #4: No

---

## [Editor Report · Acceptance letter]

5 Mar 2021

PONE-D-20-15042R1 

Economic violence against women: a case in Turkey 

Dear Dr. Alkan:

I'm pleased to inform you that your manuscript has been deemed suitable for publication in PLOS ONE. Congratulations! Your manuscript is now with our production department. 

Kind regards, 

on behalf of

Dr. Sisi Zhang 

Academic Editor

PLOS ONE